# Novel Gene Signatures as Prognostic Biomarkers for Predicting the Recurrence of Hepatocellular Carcinoma

**DOI:** 10.3390/cancers14040865

**Published:** 2022-02-09

**Authors:** Ju A Son, Hye Ri Ahn, Donglim You, Geum Ok Baek, Moon Gyong Yoon, Jung Hwan Yoon, Hyo Jung Cho, Soon Sun Kim, Suk Woo Nam, Jung Woo Eun, Jae Youn Cheong

**Affiliations:** 1Department of Gastroenterology, School of Medicine, Ajou University, Suwon 16499, Korea; gracia1429@ajou.ac.kr (J.A.S.); rhkwp37@naver.com (H.R.A.); diane885@naver.com (D.Y.); ptok99@hanmail.net (G.O.B.); ymk8028@hanmail.net (M.G.Y.); pilgrim8107@hanmail.net (H.J.C.); cocorico99@gmail.com (S.S.K.); 2Department of Biomedical Sciences, Graduate School of Medicine, Ajou University, Suwon 16499, Korea; 3Department of Pathology, College of Medicine, The Catholic University of Korea, Seoul 06591, Korea; yjh0875@hanmail.net (J.H.Y.); swnam@catholic.ac.kr (S.W.N.); 4Functional RNomics Research Center, The Catholic University of Korea, Seoul 06591, Korea

**Keywords:** biomarker, prognosis, hepatocellular carcinoma, recurrence

## Abstract

**Simple Summary:**

A high percentage of patients who undergo surgical resection for hepatocellular carcinoma (HCC) experience recurrence. Therefore, identification of accurate molecular markers for predicting recurrence of HCC is important. We analyzed recurrence and non-recurrence HCC tissues using two public omics datasets comprising microarray and RNA-sequencing and found novel gene signatures associated with recurrent HCC. These molecules might be used to not only predict for recurrence of HCC but also act as potential prognostic indicators for patients with HCC.

**Abstract:**

Hepatocellular carcinoma (HCC) has a high rate of cancer recurrence (up to 70%) in patients who undergo surgical resection. We investigated prognostic gene signatures for predicting HCC recurrence using in silico gene expression analysis. Recurrence-associated gene candidates were chosen by a comparative analysis of gene expression profiles from two independent whole-transcriptome datasets in patients with HCC who underwent surgical resection. Five promising candidate genes, *CETN2*, *HMGA1*, *MPZL1*, *RACGAP1*, and *SNRPB* were identified, and the expression of these genes was evaluated using quantitative reverse transcription PCR in the validation set (*n* = 57). The genes *CETN2*, *HMGA1*, *RACGAP1*, and *SNRPB*, but not *MPZL1*, were upregulated in patients with recurrent HCC. In addition, the combination of *HMGA1* and *MPZL1* demonstrated the best area under the curve (0.807, 95% confidence interval [CI] = 0.681–0.899) for predicting HCC recurrence. In terms of clinicopathological correlation, *CETN2*, *MPZL1*, *RACGAP1*, and *SNRPB* were upregulated in patients with microvascular invasion, and the expression of *MPZL1* and *SNRPB* was increased in proportion to the Edmonson tumor differentiation grade. Additionally, overexpression of *CETN2*, *HMGA1*, and *RACGAP1* correlated with poor overall survival (OS) and disease-free survival (DFS) in the validation set. Finally, Cox regression analysis showed that the expression of serum alpha-fetoprotein and *RACGAP1* significantly affected OS, whereas platelet count, microvascular invasion, and *HMGA1* expression significantly affected DFS. In conclusion, *HMGA1* and *RACGAP1* may be potential prognostic biomarkers for predicting the recurrence of HCC after surgical resection.

## 1. Introduction

Hepatocellular carcinoma (HCC) is the most common type of primary liver cancer and the third leading cause of cancer-related mortality worldwide [1]. Immunotherapy is a systemic treatment option, with immune checkpoint inhibitors such as tyrosine kinase inhibitors (e.g., lenvatinib, cabozantinib, regorafenib) and anti-angiogenic monoclonal antibodies (ramucirumab) [2,3,4]; however, the response to ICIs is only 20–30% of patients [5]. Surgical resection is a curative treatment option which is currently chosen first for HCC [6]; however, up to 70% of patients who undergo surgical resection may experience recurrence of HCC, which can affect their long-term prognosis [7]. Thus, it is important to identify molecular markers that accurately predict recurrence and prognosis after HCC resection.

RNA sequencing (RNA-seq) and microarray are two of the most commonly used high-throughput technologies for transcriptome profiling, and characteristic molecular gene signatures identified by these techniques could potentially be used as biomarkers to predict the recurrence of HCC. In recent studies, Wang et al. reported that *NUF2* is related to the recurrence of HCC, and that higher *NUF2* expression is associated with poor overall survival (OS) and recurrence-free survival in patients with HCC [8]. Another study showed that *PCR1* promotes early recurrence of HCC and is associated with the Wnt/β-catenin signaling pathway [9]. Several researchers have studied integrated models comprising multiple genes. A five-gene signature (including *FKBP11*, *SCRIB*, *SLC38A2*, *SORBS2*, and *STAB2*) has been reported as a predictive marker for recurrence-free survival of HCC [10]. In addition, a long noncoding RNA (lncRNA) signature consisting of six lncRNAs (*MSC-AS1*, *POLR2J4*, *EIF3J-AS1*, *SERHL*, *RMST*, and *PVT1*) was significantly associated with recurrence-free survival of HCC in the TCGA cohort [11].

Although HCC recurrence has been studied extensively, the underlying molecular mechanism of recurrent HCC remains largely unknown. The development of a prediction marker for HCC recurrence after treatment is fundamental for planning a therapeutic strategy and increasing the life expectancy of patients with HCC. In this study, we evaluated differentially expressed genes in the HCC tissues of patients who underwent surgical resection using two datasets consisting of independent transcriptome data analyzed by microarray and RNA-seq, respectively. To develop predictive markers for HCC recurrence, we analyzed their relevance to HCC recurrence using gene set enrichment analysis (GSEA) to compare them with genes from four other datasets and validated them using GSE39791, which is composed of data from tumor and matching non-tumor surrounding tissues of 72 patients with HCC who underwent surgical resection as the primary treatment.

## 2. Materials and Methods

### 2.1. The Integrative Analysis Strategy of Microarray and RNA-Seq Datasets

The strategy used in the present study is represented in Figure 1A. In this study previously published RNA microarray data (GSE89377) and RNA-seq data (GSE114564) has been analyzed using a total of 85 HCC tissue samples from patients with recurrent and non-recurrent HCC who underwent surgical resection. The clinical information of patients and datasets were acquired from corresponding authors upon our request. The processed microarray data from GSE89377 and the processed RNA-seq data from GSE114564 were downloaded as FPKM values and used in this study. The GSE89377 consisted of 32 HCC tissues from 24 patients with recurrent HCC and 8 patients with non-recurrent HCC. The GSE114564 consisted of 53 HCC tissues from 21 patients with recurrent HCC and 32 patients with non-recurrent HCC. Those two databases were used to identify mainly overexpressed gene candidates in HCC. The GSEA was used to validate candidate genes associated with HCC recurrence. The GSE39791 [12] was used to evaluate core gene signatures related to HCC recurrence.

### 2.2. Data Analysis

In this study, we requested the material and methods for processing raw data of RNA microarray and RNA-seq from the corresponding authors.

Microarray data analysis for whole transcriptome from HCC patients was performed using the following software: GenomeStudio (version 3.0, Illumina), GenPlex™ (version 3.0, ISTECH, Inc., Seoul, Korea), EXCEL (Microsoft), and GSEA (version 2.07, Broad Institute). Briefly, GenomeStudio (version 3.0) was used for the data acquisition and calculation of signal values on Illumina expression beadchip. Normalization of expression data and hierarchical clustering was performed by GenPlex™ (version 3.0). For primary data filtering, spots with a P-call (Detection call *p*-value < 0.1) were selected and normalized via quantile normalization. A multitude of analyses were performed using the normalized and filtered data. The primary microarray data are available in the Gene Expression Omnibus (GEO) database (GSE89377) [13].

The RNA-sequencing was performed on Illumina HiSeq2000 machines (Illumina) using the standard Illumina RNA-seq protocol with a read length of 2 × 100 bases (paired-end). At first, the quality of RNA-seq data for each sample was assessed using FastQC and raw reads were trimmed using cutadapt to get high-quality reads with a minimum Phred quality value of 30. Processed reads of all samples were aligned to the hg19 human reference genome using TopHat. Cufflinks was used for transcript abundance estimation with performing reference annotation based transcript (RABT) assembly to measure the level of gene expression. The assemblies were produced separately for each sample, and CuffMerge was used to merge assemblies and construct consensus transcripts with the reference genome annotation. Cuffquant was utilized to quantify the gene-level of expression profile for downstream analysis processes. The data was uploaded in the GEO database (Accession Number: GSE114564). The article originally describing these datasets is not published yet.

### 2.3. Clinical Characteristics of Patients

Fifty-seven pairs of HCC and corresponding non-tumor tissue samples from patients who underwent surgical resection between January 2014 and December 2018 were obtained from the Biobank of Ajou University Hospital, a member of the Korea Biobank Network. All patients who participated in the study provided written informed consent to publish the results according to the Declaration of Helsinki. Tissue samples were immediately frozen in liquid nitrogen. The clinical characteristics of the patients are shown in Appendix A. The study protocol was approved by the Institutional Review Board of Ajou University Hospital (AJRIB-BMT-KSP-16-365 and AJIRB-BMR-SMP-17-189).

### 2.4. Analysis of Differentially Expressed Genes (DEGs)

For analysis of differentially expressed genes between the recurrent and the non-recurrent groups, Significance Analysis of Microarrays (SAM) was used for each dataset. At first, the microarray data were analyzed using the samr package in R, a free software environment available at http://www.r-project.org/, accessed on 13 August 2018. The statistical significance of differential gene expression was determined by assimilating a set of gene-specific *t*-tests. The scores for each gene were assigned based on changes in gene expression relative to the standard deviation of repeated measurements for that gene. Genes with scores higher than the threshold were considered significant. The percentage of genes with scores that fall within a window around the score identified by chance is the false discovery rate (FDR) [14]. In case of RNA-seq analyses, we used preprocessed gene abundances normalized by library and gene length by calculating fragments per kilobase of exon per million mapped reads (FPKM). For RNA-seq FPKM datasets, we also used “SAMSeq” in samr package in R [15]. In particular, the “assay.type” option was selected as “seq” to perform analysis suitable for sequence data. The identification of differences in gene expression in RNA-seq data was analyzed in the same way as microarray data.

### 2.5. Gene Set Enrichment Analysis

To identify the genes that were enriched from known gene sets associated with HCC recurrence, we downloaded gene sets from MSigDB (http://software.broadinstitute.org/gsea/msigdb, accessed on 13 August 2018) at the Broad Institute GSEA (http://www.broadinstitute.org/gsea, accessed on 13 August 2018).

GSEA was performed to assess the relevance of the gene sets (WOO_LIVER_CANCER_RECURRENCE_UP, YOSHIOKA_LIVER_CANCEREARLY_RECURRENCE_UP, WANG_RECURRENT_LIVER_CANCER_UP, and HOSHIDA_LIVER_CANCER_LATE_RECURRENCE_UP), which have been previously studied for HCC recurrence and to identify recurrence-related gene signatures. Given a dataset in which genes are rank-ordered by the correlation of their expression level with the phenotype of interest, a basic GSEA provides a score that quantifies the degree of enrichment of a given gene set at the top (positive correlation) or bottom (negative correlation) of the rank-ordered dataset. The normalized enrichment score (NES) was then calculated. If both the normal *p* value and FDR *q*-value were less than 0.05, the gene set was identified as significantly enriched.

### 2.6. RNA Extraction and Quantitative Reverse Transcription Polymerase Chain Reaction

Total RNA was isolated from frozen tissues using QIAzol (Qiagen, Hilden, Germany) according to the manufacturer’s instructions. To synthesize cDNA, 500 ng of total RNA was reverse transcribed using 5X PrimeScript™ RT Master Mix (Takara Bio, Otsu, Japan) in a final volume of 10 µL. Quantitative real-time polymerase chain reaction (qRT-PCR) was performed using amfiSure qGreen Q-PCR Master Mix (GenDEPOT, Barker, TX, USA) in accordance with the manufacturer’s instructions and the CFX Connect Real-Time PCR Detection System (Bio-Rad, Richmond, CA, USA). The primers used for qRT-PCR are listed in Appendix A. The results were computed using the 2^−∆∆Ct^ method, and *HMBS* was used for normalization. All measurements were performed in triplicate.

### 2.7. Statistical Analysis

Results are presented as mean ± standard deviation (SD). Statistical significance of the difference between two groups was assessed by paired Student *t*-test or unpaired Welch’s *t*-test and one-way analysis of variance (ANOVA) in GraphPad Prism version 8.0 (GraphPad Software, San Diego, CA, USA). Kaplan–Meier survival curves with the log-rank test were performed to assess the significant prognostic power between the two patient groups. IBM SPSS software (IBM SPSS Statistics for Windows, version 22.0, released 2013, IBM Corp.) was used for receiver operating characteristic (ROC) analysis and Cox proportional hazard regression analysis. ROC curves were analyzed to evaluate sensitivity, specificity, and respective areas under the ROC (AUROCs) with 95% confidence intervals (CIs) for each candidate biomarker. Univariate Cox regression and multivariate Cox regression analyses were conducted to evaluate the independent prognostic value of the signature using the “survival” R package. Hazard proportional assumption of Cox regression model was confirmed by using the loglikelihood test. To adjust the interaction between variables, multivariate analysis was performed. Variables with *p* < 0.05 in the univariate Cox regression were included in the multivariate Cox regression analysis.

## 3. Results

### 3.1. Gene Selection for HCC Recurrence Using Systemic Analysis of Tissue-Based Microarray and RNA-Seq Dataset

To identify novel gene signatures for detecting recurrence in HCC patients who underwent surgical resection, we analyzed gene expression profiles using two different datasets analyzed by microarray and RNA-seq. We requested clinical information of patients and datasets from corresponding authors to perform the integrative analysis. The workflow of this study is shown in Figure 1A. Microarray data (GSE89377) were acquired from 32 tissues (8 non-recurrence tissues and 24 recurrence tissues), and RNA-seq data (GSE114564) were obtained from 53 tissues (32 non-recurrence tissues and 21 recurrence tissues). Upregulated and downregulated genes in the recurrence group of each dataset were estimated using SAM, a supervised learning software for genomic expression data mining (Figure 1B). A total of 2385 and 5927 genes were differentially expressed in the two datasets, and a Venn diagram analysis of the two sets revealed that 981 genes that were differentially expressed were part of a common molecular signature for both datasets (Figure 1C). We found that the 981 differentially expressed genes (DEGs) were significantly correlated with microarray and RNA-seq data (Pearson’s correlation coefficient *r* = 0.9178, *p* < 0.001) (Figure 1D). Gene ontology (GO) analysis demonstrated that 981 DEGs were associated with HCC biological processes such as the Wnt signaling pathway, angiogenesis, and blood coagulation (Figure 1E).

### 3.2. Identification of Predictive Molecular Signature for HCC Recurrence Using the Public Database

Next, to evaluate the biological relevance of the recurrence-associated molecular signature, we performed GSEA with the 981 highly correlated genes from microarray and RNA-seq datasets in four recurrence-related gene sets (Figure 2A). We found that a total of 19 genes were enriched as core genes in the four gene sets. The gene lists and the enrichment scores are listed in Appendix A. In addition, we confirmed that all 19 core genes were significantly upregulated in microarray datasets but not all genes in the RNA-seq dataset (Figure 2B).

To clarify that 19 core genes were closely related to the clinical outcome of HCC, we used GEO (GSE39791) and TCGA_LIHC databases. In GSE39791, 17 of the 19 core genes were upregulated in HCC tissues compared to adjacent non-tumor tissues. The two core genes that were not upregulated were *ALDOA* and *SH3GLB* (Figure 3A). To select more accurate diagnostic molecules, ROC analysis was performed. Of the 19 genes, six genes were highly sensitive and specific for biomarkers with an area under the curve (AUC) of more than 0.8 (Figure 3B). Next, we analyzed whether these six genes were associated with survival using the TCGA_LIHC database. Patients were divided into two groups, one with high expression and one with low expression for each of the six genes. The Kaplan–Meier survival curves indicated that patients with higher gene expression levels had lower disease-free survival (DFS) rates than those with lower gene expression levels, with the exception of *MYO6* (Figure 3C, *p* < 0.05). Finally, five genes, *CETN2*, *HMGA1*, *MPZL1, RACGAP1*, and *SNRPB* were selected for validation.

### 3.3. Confirmation of Five Recurrence Related Genes in Validation Set

The validation set included 57 matched pairs of human HCC tissues and adjacent non-tumor tissues. The baseline characteristics are shown in Appendix A. The expression levels of the five selected genes were evaluated in these tissues using qRT-PCR (Figure 4A). In over 80% of the tested HCC tissues, all five genes were overexpressed (*CETN2*, 80%; *HMGA1*, 93%; *MPZL1*, 89%; *RACGAP1*, 93%; and *SNRPB*, 82%). Figure 4B shows the ROC curves of the five genes for diagnosing HCC. The AUC values of the five genes were as follows: 0.701 for *CETN2* [95% CI: 0.608–0.783], 0.812 for *HMGA1* (95% CI: 0.728–0.879), 0.731 for *MPZL1* (95% CI: 0.639–0.809), 0.890 for *RACGAP1* (95% CI: 0.818–0.941), and 0.765 for *SNRPB* (95% CI: 0.676–0.839).

The validation set included 37 patients with non-recurrent HCC and 20 patients with recurrent HCC. Figure 4C shows the expression values of the five genes according to recurrence state. With the exception of *MPZL1*, the genes were significantly upregulated in patients with recurrent HCC compared to expression levels in patients with non-recurrent HCC. Next, we evaluated the prognostic performance of the five selected genes in recurrent HCC compared to serum alpha-fectoprotein (AFP) using AUROC. The AUC of serum AFP for predicting HCC recurrence was 0.628, and the AUC of *CETN2*, *HMGA1*, *MPZL1*, *RACGAP1*, and *SNRPB* were 0.689, 0.755, 0.524, 0.754, and 0.677, respectively. No single gene was found to be superior to serum AFP for predicting tumor recurrence (Figure 4D; left, Appendix A).

To acquire the most effective predictive panel of genes, we evaluated the predictive capability of different combinations of the five genes and serum AFP levels. Figure 4D shows a comparison of ROC curves of the derived gene panels and serum AFP. The combinations of each of the five genes and serum AFP showed no significant differences in AUROC compared to serum AFP alone (Figure 4D; middle). On the other hand, for the combination of the five genes, *HMGA1* combined with *MPLZ1* produced the most accurate results for predicting HCC recurrence (AUC = 0.807, 95% CI = 0.681–0.899) (Figure 4D; right). The AUC value, 95% CI, and *p* value of each factor are listed in Appendix A.

### 3.4. Prognostic Implication of the Candidate Biomarker Genes

We evaluated whether the five selected genes were related to the clinicopathological characteristics of patients with HCC in the validation set: *CETN2*, *MPZL1*, *RACGAP1*, and *SNRPB* were significantly overexpressed in patients with vascular invasion, but *HMGA1* was not overexpressed (Figure 5A). The five selected genes were upregulated in proportion to Edmondson–Steiner tumor differentiation grades, but only *MPZL1* and *SNRPB* showed statistical significance (Figure 5B). Next, we examined the prognostic impact of these five genes in the validation set. In patients with HCC, the, high expression levels of *CETN2*, *HMGA1*, and *RACGAP1* were significantly associated with OS and DFS (Figure 5C).

To further validate the prognostic factors affecting OS and DFS, univariate and multivariate Cox regression analyses were performed. Univariate analysis showed that total bilirubin (*p* = 0.015), creatinine (*p* = 0.043), serum alanine aminotransferase (ALT) (*p* = 0.035), serum AFP (*p* < 0.001), protein induced by vitamin K absence-II (PIVKA II) (*p* = 0.003), advanced tumor stage (modified UICC stage III-IV, *p* = 0.008), *SNRPB* expression (*p* = 0.044), *CETN2* expression (*p* = 0.020), *HMGA1* expression (*p* = 0.020), and *RACGAP1* expression (*p* < 0.001) were significantly associated with OS. In addition, platelet (*p* = 0.015), serum AFP (*p* = 0.003), PIVKA II (*p* = 0.002), presence of microvascular invasion (MVI; *p* < 0.001), advanced tumor stage (mUICC stage III-IV, *p* < 0.001), *SNRPB* expression (*p* = 0.008), *CETN2* expression (*p* = 0.008), *HMGA1* expression (*p* = 0.001), and *RACGAP1* expression (*p* < 0.001) were significantly associated with DFS (Table 1).

Multivariate Cox regression analysis was used to determine the significant predictors of OS and DFS. The prognostic factors for OS were serum AFP (hazard ratio [HR] = 1.000, 95% CI = 1.000–1.000, *p* = 0.003) and *RACGAP1* expression (HR = 34.162, 95% CI = 3.486–334.737, *p* = 0.002). The prognostic factors for DFS were platelet (HR = 1.015, 95% CI = 1.005–1.024, *p* = 0.002), presence of MVI (HR = 18.674, 95% CI = 4.103–84.996, *p* < 0.001), and *HMGA1* expression (HR = 2.905, 95% CI = 1.148–7.353, *p* = 0.024) (Table 1).

## 4. Discussion

The present study compared gene expression between recurrent HCC tissues and non-recurrent HCC tissues after surgical resection with integrative analysis using two different datasets (GSE89377 and GSE114564) analyzed by microarray and RNA-seq, respectively. Microarray and RNA-seq are two of the most commonly used high-throughput technologies for gene expression profiling. However, little is known about the reproducibility of these results. Therefore, we calculated the correlation between microarray and RNA-seq results to explore the reproducibility between the two techniques and to find more precise markers.

We performed GSEA with four different gene sets that were previously studied for HCC recurrence to identify the relevance of the verified genes with recurrence. For more accurate verification, we used the GSE39791 database to evaluate the expression of 19 genes to confirm that they were upregulated in HCC tissues. We identified five genes (*CETN2*, *HMGA1*, *MPZL1*, *RACGAP1*, and *SNRPB*) associated with HCC prognosis that are potential biomarkers for HCC recurrence. Among the five genes, *CETN2*, *HMGA1*, *RACGAP1*, and *SNRPB* were significantly upregulated in patients with recurrent HCC. Additionally, the combination of *HMGA1* and *MPZL1* had the best AUC for predicting HCC recurrence.

In terms of clinicopathological correlation and prognostic implication, *CETN2*, *MPZL1*, *RACGAP1*, and *SNRPB* were overexpressed in patients with vascular invasion. In addition, the expression levels of *MPZL1* and *SNRPB* were significantly upregulated in proportion to the Edmonson grade. Serum AFP level and *RACGAP1* were significantly correlated with OS; and, platelets and the presence of MVI and *HMGA1* were associated with DFS in patients with HCC.

A member of the centrin protein family, *CETN2* is a small, evolutionarily conserved, calcium-binding protein that plays an important role in humans [16,17]. Centrins are localized to centrosomes and contain two pairs of EF-hands, helix-loop-helix structures in which calcium can bind, separated by a linker region [18,19]. Only a few studies have revealed a link between *CETN2* and cancer. To our knowledge, this is the first study to demonstrate that *CETN2* is upregulated in HCC tissues compared to that in non-HCC tissues. In addition, high expression of *CETN2* was related to poor prognosis of patients with HCC in the TCGA_LIHC database and validation set. Furthermore, overexpression of *CETN2* was confirmed in recurrent HCC tissues in the validation set.

Located on chromosome 6p21, *HMGA1* is a high-mobility group A1 protein. It has three isoforms due to alternative splicing: *HMGA1a*, *HMGA1b*, and *HMGA1c* [20]. Although the HMGA family does not have intrinsic transcriptional activity, it is involved in the remodeling of chromatin structure, and it regulates the transcriptional activity of an increasing number of genes [21]. High expression of *HMGA1* has been reported to play an important role in embryonic development [22,23]. Recent studies have revealed that *HMGA1* is associated with various types of cancers including breast [24,25], lung [26,27], and colorectal [28,29] cancer types. Although several studies have shown that *HMGA1* plays an important role in the development of HCC [30,31], only one study has reported that the recurrence rate of HCC correlated with high *HMGA1* mRNA expression and that it could be used as a prognostic marker for HCC [32]. In the present study, we evaluated the potential of *HMGA1* as a prognostic marker. Cox proportional hazard regression analysis demonstrated that *HMGA1* was associated with decreased DFS in patients with HCC (HR = 2.905, 95% CI = 1.148–7.353, *p* = 0.024).

*A* member of the immunoglobulin superfamily, *MPZL1* is composed of an extracellular segment and an intracellular portion with two immunoreceptor tyrosine-based inhibitory motifs (ITIMs) [33]. The phosphorylated ITIMs specially bind to the SH2 domain [34,35]. Although *MPZL1* is overexpressed in malignant tumor tissues [36,37,38,39], *MPZL1* expression and its relationship with HCC prognosis and recurrence have not been evaluated. This study is the first to report that an increased expression of MPZL1 is associated with poor prognosis of HCC. According to the Edmondson grade, the expression of MPZL1 was increased in patients with invasion although MPZL1 expression was not significantly different between recurrent HCC patients and non-recurrent HCC patients. These results support the view that the combination of HMGA1 and MPZL1 may be a potential biomarker as they act in a complementary way, predicting recurrent HCC with an AUC of 0.807 (95% CI = 0.681–0.899).

A number of studies have identified the underlying mechanisms of *RACGAP1* upregulation in HCC: ECT2 interacts and co-localizes with RACGAP1 and protects RACGAP1 from degradation; and, RACGAP1 promotes ECT2-mediated RhoA activation and metastasis in HCC cells [40]. Wang et al. showed that the pseudogene *RACGAP1P* activates *RACGAP1*, and miR-15-5p suppresses the expression of both *RACGAP1P* and *RACGAP1* [41]. The lncRNA *MAGI2-AS3* interacts with KDM1A and promotes histone demethylation of histone H3 lysine 4 (H3K4me2) in the *RACGAP1* promoter region, and silencing *RACGAP1* inhibits tumor growth in vivo [42]. Regarding HCC prognosis, there is one study that showed that upregulation of *RACGAP1* is associated with early recurrence of HCC [43]. In the present study, we further showed that high *RACGAP1* expression could be an independent prognostic factor for HCC by Cox proportional hazard regression analysis (HR = 34.162, 95% CI = 3.486–334.737, *p* = 0.002).

Overexpression of *SNRPB* has been reported to be associated with poor prognosis in patients with HCC. It has also been suggested that c-Myc mediates upregulation of *SNRPB* [44]. The SNRPB protein is a key component of the spliceosome, which is responsible for alternative splicing regulation in HCC. Additionally, SNRPB-mediated RNA splicing promotes HCC cell proliferation and it is correlated with the maintenance of cell stemness in HCC cells [45]. Our study presents the first evidence of a relationship between SNRPB and HCC recurrence.

It remains unclear whether expression of specific genes in HCC tissue could be an independent predictor of recurrence. To identify recurrence-related genes in HCC, we proposed a detailed methodology through combinatorial analysis between two different platforms, microarrays, and RNA-seq datasets. As a result, CETN2, HMGA1, MPZL1, RACGAP1, and SNRPB were identified as potential predictive biomarkers of recurrent HCC. Indeed, we have demonstrated that the combination of HMGA1 and MPZL1 has excellent prognostic value for distinguishing patients at high risk of HCC recurrence in the validation cohort. These patients at high risk of recurrence may achieve better clinical outcome by active change of their post-treatment surveillance intervals and/or the adjuvant treatment to prevent recurrence. In the future, more accurate clinical risk prediction model for HCC patients at high risk of recurrence would be derived by integrating the expression of recurrence-related genes and various clinical characteristics; and, the present study would serve as a cornerstone for deriving the more accurate risk prediction model for HCC patients.

This study has some limitations. First, the number of patients enrolled in the validation study was relatively small. Besides, the validation study was designed as a retrospective study, and it was not performed in an external cohort. Thus, to integrate our results into real clinical practice to discriminate patients at high risk of recurrence, rigorous external validation in a larger prospective cohort should be performed. Second, we did not perform the mechanism studies underlying the promotion of HCC recurrence by the recurrence related genes. Among the five genes, the mechanism of *HMGA1*, *MPZL1*, *RACGAP1*, and *SNRPB* in cancer progression were reported in several prior studies. However, the mechanism of *CETN2* action in cancers, specifically HCC, was not identified. Further studies are required to evaluate the underlying mechanism of *CETN2* in patients with HCC.

## 5. Conclusions

In conclusion, the present study derived potential molecular biomarkers for recurrent HCC using a systematic, genome-wide discovery approach. The following were significantly overexpressed in patients with HCC and were significantly correlated with HCC recurrence: *CETN2, HMGA1, MPZL1, RACGAP1,* and *SNRPB*. Furthermore, the combination of *HMGA1* and *MPZL1* showed excellent predictive capabilities for HCC recurrence. Moreover, *HMGA1* and *RACGAP1* could be used as independent prognostic indicators in patients with HCC. The integrative analysis that we used in this study could be used to derive genes that could be used as potential biomarkers for HCC recurrence, and these genes may help identify patients who need to be followed closely to lower the recurrence rate after liver resection.

## Figures and Tables

**Figure 1 cancers-14-00865-f001:**
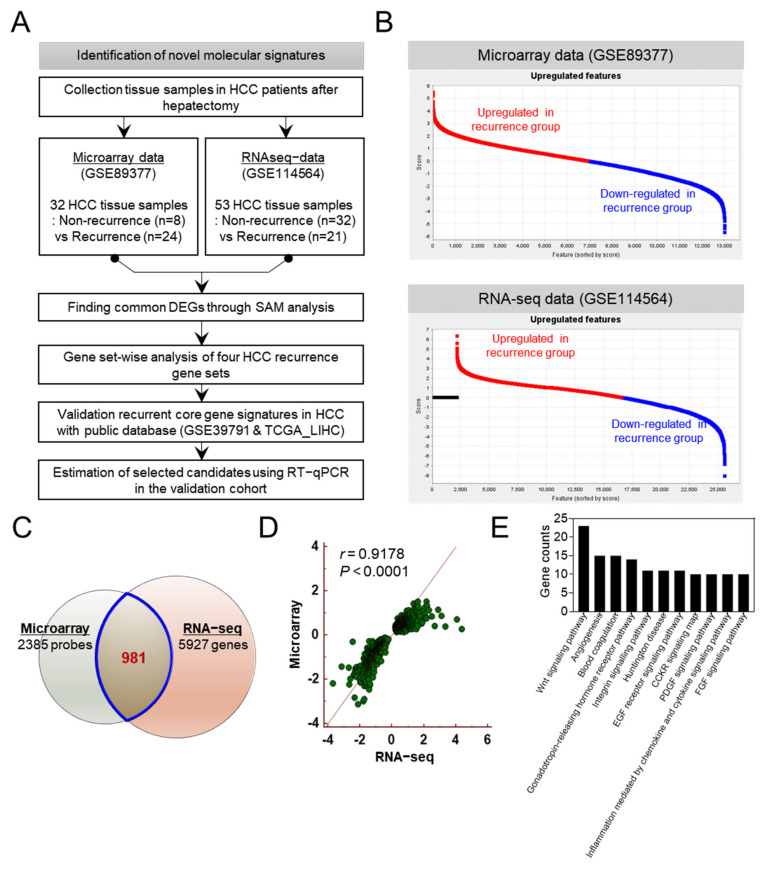
Integrative analysis of tissue−based microarray data and RNA−seq data to identify novel gene signatures for the recurrence of hepatocellular carcinoma (HCC). (**A**) Flow chart demonstrating the methodology used to identify gene signatures for predicting the recurrence of hepatocellular carcinoma. (**B**) Gene expression plot of each dataset displaying upregulated and downregulated genes. (**C**) Venn diagram analysis to determine common differentially expressed genes in HCC tissues identified using two different datasets. (**D**) Scatter plot of the correlation between microarray and RNA−seq results for 981 differentially expressed genes (DEGs; *r* = 0.9178, *p* < 0.001). (**E**) The analysis of gene ontology revealed significant enrichment of differentially expressed gene signatures associated with biological processes.

**Figure 2 cancers-14-00865-f002:**
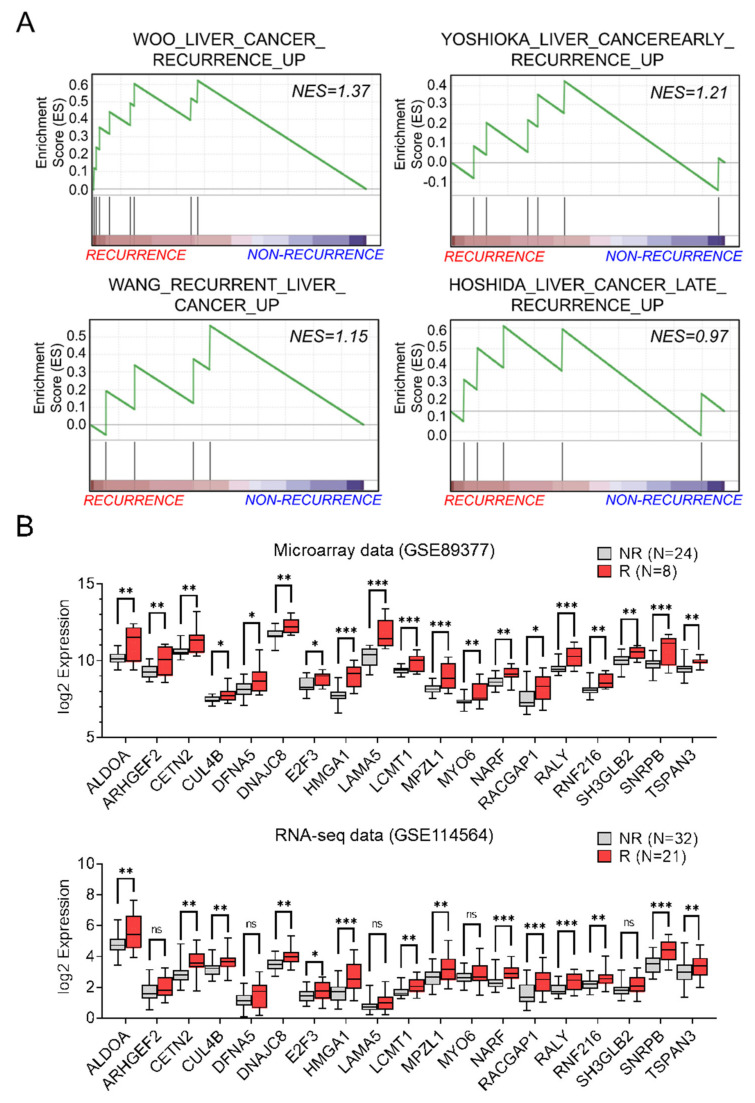
Nineteen core genes were associated with recurrence and upregulated in patients with recurrent HCC. (**A**) Correlation between gene set enrichment analysis (GSEA) of the 981 DEGs and HCC recurrence in four datasets. (**B**) Expression level of 19 genes in recurrence and non−recurrence tissues in both datasets. Statistically significant differences were determined using the Unpaired *t*−test (* *p* < 0.05, ** *p* < 0.01, *** *p* < 0.001). ns; not significant.

**Figure 3 cancers-14-00865-f003:**
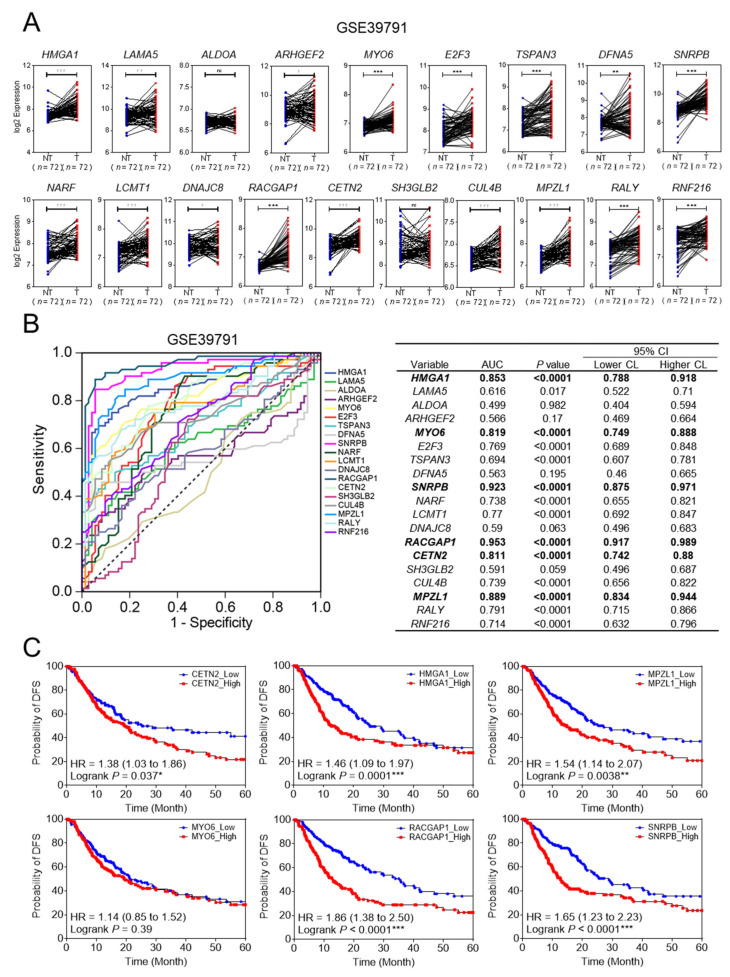
Validation of the final five recurrence-related genes in patients with HCC using GEO database and TCGA data. (**A**) Gene expression data shows the relative expression of the 19 genes in the GEO database (GSE39791). (Welch’s *t*-test; * *p* < 0.05, ** *p* < 0.01, *** *p* < 0.001) (**B**) Receiving operating curves of the 19 genes with area under the curve (AUC) > 0.8 for predicting HCC recurrence in the GEO database (GSE39791). (**C**) Kaplan–Meier survival analysis of disease-free survival based on the expression of six genes in the TCGA_LIHC database. Hazard ratios (HR) with 95% confidence intervals and *p* values were calculated using the log-rank test, * *p* < 0.05, ** *p* < 0.01, *** *p* < 0.001.

**Figure 4 cancers-14-00865-f004:**
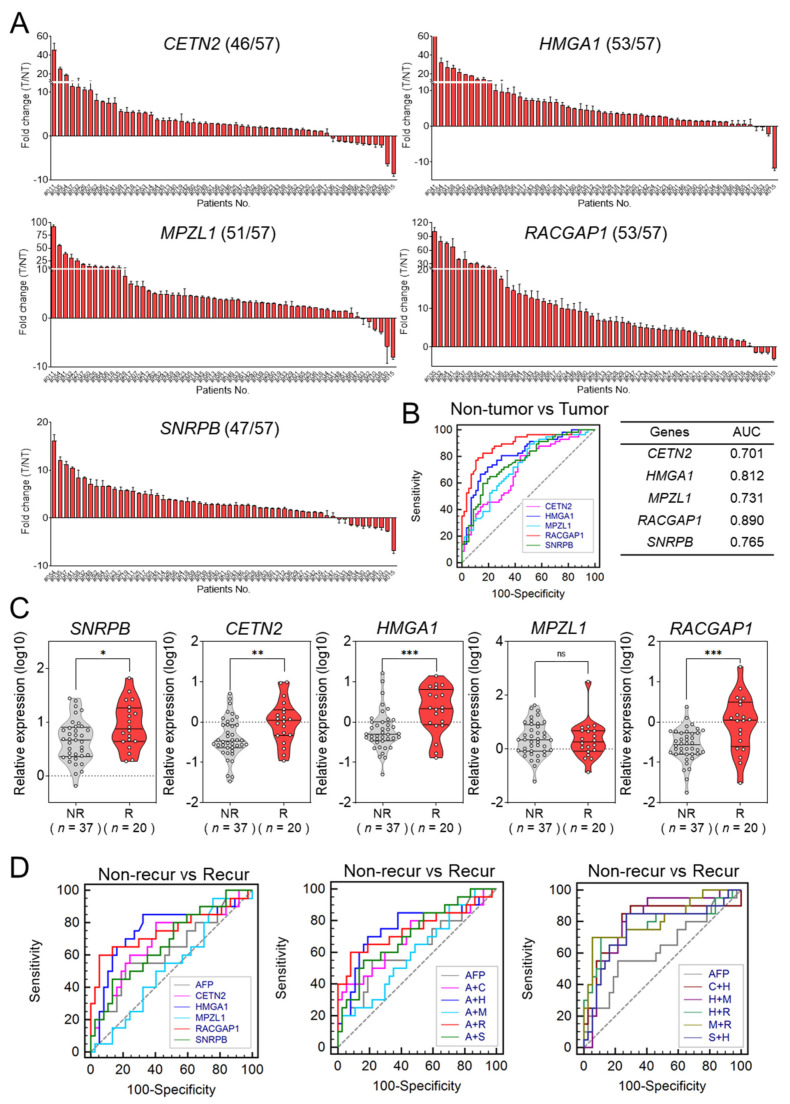
Expression levels of the selected five genes in HCC tissues and their diagnostic value for predicting the recurrence of HCC in the validation set. (**A**) Gene expression levels of the five genes in 57 matched pairs of human HCC tissues and adjacent non−tumor tissues. (**B**) Receiver operating characteristic (ROC) curve analysis of the five genes for predicting the recurrence of HCC in the validation set. (**C**) Relative expression of the final five genes according to the recurrence status of HCC in the validation set (Welch’s *t*−test; * *p* < 0.05, ** *p* < 0.01, *** *p* < 0.001) (**D**) AUCs for each of the five core genes (**left**), the combination of AFP with the five genes (**middle**), and the combination of two gene signatures (**right**) for predicting the recurrence of HCC.

**Figure 5 cancers-14-00865-f005:**
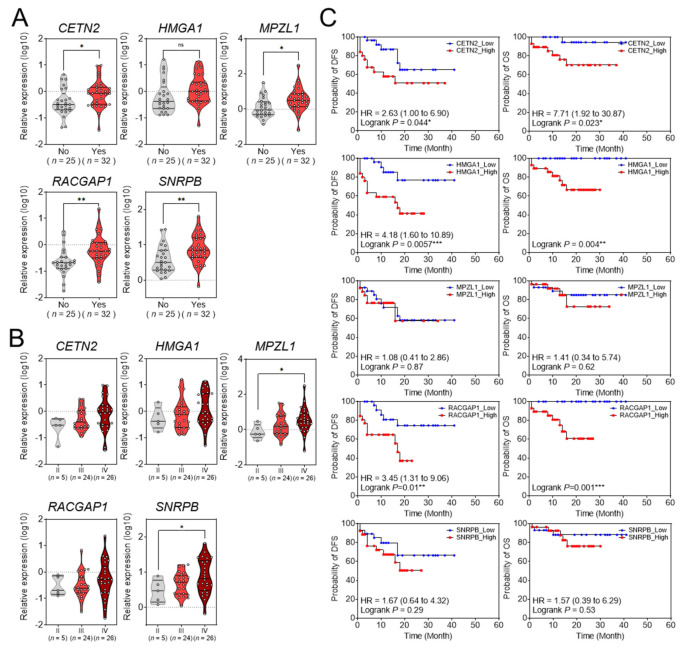
The relationship between gene expression and clinicopathological characteristics. (**A**) Relative gene expression in HCC tissues with or without vascular invasion. * *p* < 0.05, ** *p* < 0.01, *** *p* < 0.001. (**B**) Relative expression of the five genes according to the modified UICC stage in the validation set. * *p* < 0.05. (**C**) Kaplan−Meier survival curve of overall survival and disease−free survival based on the expression levels of the five genes in the validation set.

**Table 1 cancers-14-00865-t001:** Univariate and multivariate Cox regression analysis of factors associated with overall survival and disease-free survival.

	OS						DFS					
	Univariate			Multivariate			Univariate			Multivariate		
Factor	HR	95% CI	*p* Value	HR	95% CI	*p* Value	HR	95% CI	*p* Value	HR	95% CI	*p* Value
**Sex (male)**	1.265	0.254–6.300	0.774				1.693	0.496–5.786	0.401			
**Age (years)**	1.008	0.941–1.080	0.815				1.028	0.981–1.076	0.249			
**Liver cirrhosis** **(yes vs. no)**	3.747	0.450–31.170	0.222				0.747	0.280–1.994	0.560			
**Platelet (10^9^/L)**	1.002	0.993–1.011	0.627				**1.007**	**1.001–1.012**	**0.015**	**1.015**	**1.005** **–** **1.024**	**0.002**
**Albumin (g/dL)**	0.685	0.323–1.451	0.323				0.713	0.464–1.096	0.123			
**Total bilirubin (mg/dL)**	**1.502**	**1.083–2.083**	**0.015**				1.380	0.993–1.917	0.055			
**Creatinine (mg/dL)**	**0.005**	**0.000–0.838**	**0.043**				0.187	0.013–2.608	0.212			
**Serum AST** **(U/L)**	1.002	0.996–1.007	0.588				1.001	0.996–1.006	0.790			
**Serum ALT** **(U/L)**	**1.007**	**1.000–1.013**	**0.035**				1.005	1.000–1.010	0.057			
**Serum AFP (ng/mL)**	**1.000**	**1.000–1.000**	**<0.001**	**1.000**	**1.000** **–** **1.000**	**0.003**	**1.000**	**1.000–1.000**	**0.003**			
**PIVKA II (mAU/mL)**	**1.000**	**1.000–1.000**	**0.003**				**1.000**	**1.000–1.000**	**0.002**			
**MVI** **(yes vs no)**	0.002	0.000–7.853	0.144				**13.464**	**4.210–43.055**	**<0.001**	**18.674**	**4.103** **–** **84.996**	**<0.001**
**mUICC** **(III-IV vs. I-II)**	**17.153**	**2.092–140.608**	**0.008**				**7.691**	**2.889–20.473**	**<0.001**			
**Tumor grade** **(3–4 vs. 1–2)**	0.045	0.000–32,060.773	0.653				22.504	0.004–144,372.189	0.486			
** *SNRPB* **	**5.672**	**1.051–30.609**	**0.044**				**4.550**	**1.496–13.836**	**0.008**			
** *CETN2* **	**4.742**	**1.280–17.572**	**0.020**				**3.259**	**1.361–7.801**	**0.008**			
** *HMGA1* **	**4.065**	**1.244–13.288**	**0.020**				**3.166**	**1.562–6.415**	**0.001**	**2.905**	**1.148** **–** **7.353**	**0.024**
** *MPZL1* **	1.435	0.506–4.073	0.497				1.300	0.654–2.584	0.453			
** *RACGAP1* **	**6.631**	**2.375–18.519**	**<0.001**	**34.162**	**3.486** **–** **334.737**	**0.002**	**5.212**	**2.386–11.383**	**<0.001**			

## Data Availability

The microarray and RNA-seq data analyzed in this study are available in the NCBI Gene Expression Omnibus (http://www.ncbi.nlm.nih.gov/geo/, accessed on 13 August 2018) under the accession codes GSE39791, GSE89377 and GSE114564, respectively. The data sources and the handling of the publicly available datasets used in this study are described in the Materials and Methods. Further details and other data that support the findings of this study are available from the corresponding authors upon request.

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
