# Peer review of "Novel Gene Signatures as Prognostic Biomarkers for Predicting the Recurrence of Hepatocellular Carcinoma"

_cancers, 2022, doi:10.3390/cancers14040865_

Round 1

Reviewer 1 Report

Son et al. present an interesting model to predict HCC recurrence using gene expression evaluated using RNA-Seq and Microarray. The work is clearly written, presented, and is of clinical relevance. I have three comments:

  1. Looking at Figure 2B, the expression of aldolase A, LAMA5 (Experimental set I) and DNAJC8 also appears to be significant between non-recurrent and recurrent groups. Can the authors justify their exclusion from the subsequent computational and experimental (PCR) analysis?
  2. Section 3.2: "19 core genes were significantly upregulated". Do the authors mean that significance was statistically significant? If not, I suggest using another apt word. 
  3. Section 2.7: Space between "twogroups".

Author Response

We thank you for your time and consideration of our submission.

Here are our responses to the Reviewer's comment

  1.    Looking at Figure 2B, the expression of aldolase A, LAMA5 (Experimental set I) and DNAJC8 also appears to be significant between non-recurrent and recurrent groups. Can the authors justify their exclusion from the subsequent computational and experimental (PCR) analysis?

    Response: Thank you for your feedback. As your comment, LAMA5 and DNAJC8 also were upregulated recurrent group compared to non-recurrent groups. However, six genes were only highly sensitive and specific for biomarkers with an area under the curve (AUC) of more than 0.8 (Fig. 3B) in ROC analysis. Next, we analyzed the Kaplan-Meier survival curves with TCGA_LIHC dataset, and MYO6 was finally excluded (Fig. 3C, P < 0.05). Then, five genes, CETN2, HMGA1, MPZL1, RACGAP1, and SNRPB were selected for experimental (PCR) analysis. We hope that this resolves the reviewer’s comment.

    2.    Section 3.2: "19 core genes were significantly upregulated". Do the authors mean that significance was statistically significant? If not, I suggest using another apt word.

    Response: Thank you for your critical suggestion. In the original manuscript, statistical testing was not performed. Statistical testing was performed again, and we described more detail results in the result section as following. 
    : “All 19 core genes were significantly upregulated in microarray data sets, but not all genes in the RNA-seq data set.” (Line 223).
    Also, we presented statistical significance in figure 2B and figure legend. (Line 243)

3.    Section 2.7: Space between "twogroups".

Response: We thank the reviewer for correcting us. We modified the term “twogroups” to “two groups” (Line 174).

The relevant and detailed description was added to the revised manuscript, and we hope that this resolves the reviewer’s comment.

Reviewer 2 Report

Dear Authors,

thanks for submitting your research. The paper, although very technical, is well written. The conclusions are sound. Results are promising,  however currently of limited interest to the clinicians. Limitations are well explained and I agree that a large cohort is desirable.

Hopefully, papers like this will further targeted antigenic diagnostic and therapeutic options for recurrent unresectable HCC.  

Overall I have no major comments. 

Author Response

We thank you for your time and consideration on our submission.

Reviewer 3 Report

Dear Editor, thank you so much for inviting me to revise this manuscript. This study addresses a current topic.

The manuscript is quite well written and organized. English could be improved.

Figures and tables are comprehensive and clear.

The introduction explains in a clear and coherent manner the background of this study.

We suggest the following modifications:

  • Introduction section: although the authors correctly included important papers in this setting, we believe some studies should be cited within the introduction ( PMID: 34167433; PMID: 34764464 ; PMID: 33508960 ), only for a matter of consistency. We think it might be useful to introduce the topic of this interesting study.
  • Methods and Statistical Analysis: nothing to add.
  • Discussion section: Very interesting and timely discussion. Of note, the authors should expand the Discussion section, including a more personal perspective to reflect on. For example, they could answer the following questions – in order to facilitate the understanding of this complex topic to readers: what potential does this study hold? What are the knowledge gaps and how do researchers tackle them? How do you see this area unfolding in the next 5 years? We think it would be extremely interesting for the readers.

However, we think the authors should be acknowledged for their work. In fact, they correctly addressed an important topic, the methods sound good and their discussion is well balanced.

One additional little flaw: the authors could better explain the limitations of their work, in the last part of the Discussion.

We believe this article is suitable for publication in the journal although some revisions are needed. The main strengths of this paper are that it addresses an interesting and very timely question and provides a clear answer, with some limitations.

We suggest a linguistic revision and the addition of some references for a matter of consistency. Moreover, the authors should better clarify some points.

Author Response

We thank you for your time and consideration of our submission.

Here are our responses to the Reviewer's comment

1. Introduction section: although the authors correctly included important papers in this setting, we believe some studies should be cited within the introduction ( PMID: 34167433; PMID: 34764464 ; PMID: 33508960 ), only for a matter of consistency. We think it might be useful to introduce the topic of this interesting study.

Response: Thank you for the valuable comment. As your comment, we revised the first part of the introduction for better introduction for the topic of the study as following.

: Hepatocellular carcinoma (HCC) is the most common type of primary liver cancer and the third leading cause of cancer-related mortality worldwide [1]. Immunotherapy is a systemic treatment option, with immune checkpoint inhibitors such as tyrosine kinase inhibitors (e.g. lenvatinib, cabozantinib, regorafenib) and anti-angiogenic monoclonal antibodies (ramucirumab) [2-4]; however, the response to ICIs is only 20-30% of patients [5]. Surgical resection is a curative treatment option which is currently chosen first for HCC [6]; however, up to 70% of patients who undergo surgical resection may experience recurrence of HCC, which can affect their long-term prognosis [7]. Thus, it is important to identify molecular markers that accurately predict recurrence and prognosis after HCC resection. (Line 43)

2. Discussion section: Very interesting and timely discussion. Of note, the authors should expand the Discussion section, including a more personal perspective to reflect on. For example, they could answer the following questions – in order to facilitate the understanding of this complex topic to readers: what potential does this study hold? What are the knowledge gaps and how do researchers tackle them? How do you see this area unfolding in the next 5 years? We think it would be extremely interesting for the readers.

Response: We appreciated the reviewer’s valuable comment. Based on your opinion, we have additionally described a more personal perspective in the Discussion section as following.

: It remains unclear whether expression of specific genes in HCC tissue could be an independent predictor of recurrence. To identify recurrence-related genes in HCC, we proposed a detailed methodology through combinatorial analysis between two different platforms, microarrays and RNA-seq data sets. As a result, CETN2, HMGA1, MPZL1, RACGAP1, and SNRPB were identified as potential predictive biomarkers of recurrent HCC. Indeed, we have demonstrated that the combination of HMGA1 and MPZL1 has excellent prognostic value for distinguishing patients at high risk of HCC recurrence in the validation cohort. These patients at high risk of recurrence may achieve better clinical outcome by active change of their post-treatment surveillance intervals and/or the adjuvant treatment to prevent recurrence. In the future, more accurate clinical risk prediction model for HCC patients at high risk of recurrence would be derived by integrating the expression of recurrence-related genes and various clinical characteristics. And, the present study would serve as a cornerstone for deriving the more accurate risk prediction model for HCC patients. (Line 401)

3. However, we think the authors should be acknowledged for their work. In fact, they correctly addressed an important topic, the methods sound good and their discussion is well balanced.

One additional little flaw: the authors could better explain the limitations of their work, in the last part of the Discussion.

Response: Thank you for the valuable comment. As your comment, we revised the last part of the discussion for better explanation of the limitation as following.

: This study has some limitations. First, the number of patients enrolled in the validation study was relatively small. And the validation study was designed as a retrospective study and was not performed in an external cohort. Thus, to integrate our results into real clinical practice to discriminate patients at high risk of recurrence, rigorous external validation in larger prospective cohort should be performed. Second, we did not perform the mechanism studies underlying the promotion of HCC recurrence by the recurrence related genes. Among the 5 genes, the mechanism of HMGA1, MPZL1, RACGAP1, and SNRPB in cancer progression were reported in several prior studies. However, the mechanism of action of CETN2 in cancers, specifically HCC, was not identified. Further studies are required to evaluate the underlying mechanism of CETN2 in patients with HCC. (Line 415)

The relevant and detailed description was added to the revised manuscript, and we hope that this resolves the reviewer’s comment.

Reviewer 4 Report

The Authors analysed the likelihood to recur in HCC patients, using using two public omics datasets with microarray and RNA-sequencing. They proosed a novel gene signatures associated with recurrent HCC. The manuscript is of interest. Few comments:

  • How did you check for the hazard proportional assumption with the Cox regression model? Did you use the loglikelihood test? Wald test? Did you check for interaction amongst variables? Is the model adjusted for interaction.
  • Tab 1. How did you check for multiple hypotesis testing?

Author Response

We thank you for your time and consideration of our submission.

Here are our responses to the Reviewer's comment

1. How did you check for the hazard proportional assumption with the Cox regression model? Did you use the loglikelihood test? Wald test? Did you check for interaction amongst variables? Is the model adjusted for interaction.

  • Response: Thank you for the valuable comment. As your comment, we revised the Statistical analysis of Materials and Methods sections as following.

: Univariate Cox regression and multivariate Cox regression analyses were conducted to evaluate the independent prognostic value of the signature using the “survival” R package. Hazard proportional assumption of Cox regression model was confirmed by using loglikelihood test. To adjust the interaction between variables, multivariate analysis was performed. Variables with p < 0.05 in the univariate Cox regression were included in the multivariate Cox regression analysis. (Line 182)

2. Tab 1. How did you check for multiple hypotesis testing?

  • Response: Thank you for your feedback. Although the multivariate test can be performed using the vif (variance inflation factor) function in R-package, Univariate survival analysis was performed for "prognostic" relevance with each clinical characteristic, and multivariate analysis was also performed in consideration of the interaction between clinical characteristics.

The relevant and detailed description was added to the revised manuscript, and we hope that this resolves the reviewer’s comment.

Round 2

Reviewer 3 Report

The authors modified the manuscript according to our suggestions.

We recommend Acceptance in its current form.

Reviewer 4 Report

I do not have any further comment